# Deep Learning for Physical Processes: Incorporating Prior Scientific Knowledge

**Emmanuel de Bézenac**,[*] **Arthur Pajot**,[*] **Patrick Gallinari**
{emmanuel.de-bezenac, arthur.pajot, patrick.gallinari}@lip6.fr
Sorbonne Universités, UMR 7606, LIP6, F-75005 Paris, France

## Abstract

We consider the use of Deep Learning methods for modeling complex phenomena like those occurring in natural physical processes. With the large amount of data gathered on these phenomena the data intensive paradigm could begin to challenge more traditional approaches elaborated over the years in fields like maths or physics. However, despite considerable successes in a variety of application domains, the machine learning field is not yet ready to handle the level of complexity required by such problems. Using an example application, namely Sea Surface Temperature Prediction, we show how general background knowledge gained from the physics could be used as a guideline for designing efficient Deep Learning models. In order to motivate the approach and to assess its generality we demonstrate a formal link between the solution of a class of differential equations underlying a large family of physical phenomena and the proposed model. Experiments and comparison with series of baselines including a state of the art numerical approach is then provided.

## 1  Introduction

A physical process is a sustained phenomenon marked by gradual changes through a series of states occurring in the physical world. Physicists and environmental scientists attempt to model these processes in a principled way through analytic descriptions of the scientist's prior knowledge of the underlying processes. Conservation laws, physical principles or phenomenological behaviors are generally formalized using differential equations. This physical paradigm has been, and still is the main framework for modeling complex natural phenomena like e.g. those involved in climate. With the availability of very large datasets captured via different types of sensors, this physical modeling paradigm is being challenged by the statistical Machine Learning (ML) paradigm, which offers a prior-agnostic approach. However, despite impressive successes in a variety of domains as demonstrated by the deployment of Deep Learning methods in fields such as vision, language, speech, etc, the statistical approach is not yet ready to challenge the physical paradigm for modeling complex natural phenomena, or at least it has not demonstrated how to. This is a new challenge for this field and an emerging research direction in the ML community. We believe that knowledge and techniques accumulated for modeling physical processes in well developed fields such as maths or physics could be useful as a guideline to design efficient learning systems and conversely, that the ML paradigm could open new directions for modeling such complex phenomena. In this paper we then raise two issues: 1) are modern ML techniques ready to be used to model complex physical phenomena, and 2) how general knowledge gained from the physical modeling paradigm could help designing efficient ML models.

In this work, we tackle these questions by considering a specific physical modeling problem: forecasting sea surface temperature (SST). SST plays a significant role in analyzing and assessing the dynamics of weather and other biological systems. Accurately modeling and predicting such dynamics is critical in various applications such as weather forecasting, or planning of coastal activities. Since 1982, weather satellites have made huge quantities of very high resolution SST data available Bernstein (1982). Standard physical methods for forecasting SST use coupled ocean-atmosphere prediction systems, based on the Navier Stokes equations. These models rely on multiple physical

---

[*]equal contribution

hypotheses and do not optimally exploit the information available in the data. On the other hand, despite the availability of large amounts of data, direct applications of ML methods do not lead to competitive state of the art results, as will be seen in section 4.

We use SST as a typical and representative problem of intermediate complexity. Our goal is not to offer one more solution to this problem, but to use it as an illustration for advancing on the challenges mentioned above. The way we handle this problem is general enough to be transfered to a more general class of transport problems.

We propose a Deep Neural Network (NN) model, inspired from general physical motivations which offers a new approach for solving this family of problems. We first motivate our approach by introducing in section 2 the solution of a general class of partial differential equations (PDE) which is a core component of a large family of transport and propagation phenomena in physics. This general solution is used as a guideline for introducing a Deep Learning architecture for SST prediction which is described in section 3. Experiments and comparison with a series of baselines is introduced in section 4. A review of related work is finally presented in section 5.

The main contributions of this work are: 1) an example showing how to incorporate general physical background for designing a NN aimed at modeling a relatively complex prediction task. We believe the approach to be general enough to be used for a family of transport problems obeying general advection-diffusion principles. 2) formal links between our model's prediction and the solution of a general advection diffusion PDE 3) an unsupervised model for estimating motion fields, given a sequence of images. 4) a proof, on a relatively complex physical modeling problem, that full data intensive approaches based on deep architectures can be competitive with state of the art dedicated numerical methods.

## 2    PHYSICAL MOTIVATION

Forecasting consists in predicting future temperature maps using past records. Temperatures are acquired via satellite imagery. If we focus on a specific area, we can formulate the problem as prediction of future temperature images of this area using past images. The classical approach to forecasting SST consists in using numerical models representing prior knowledge on the conservation laws and physical principles, which take the form of PDEs. These models are then coupled with SST data using assimilation techniques in order to adjust to initial conditions. It is then integrated forward in time to predict SST evolution. For the sea surface, temperature variation is related to a fluid transport problem. In fluids, transport occurs through the combination of two principles: *advection* and *diffusion*. During *advection*, a fluid transports some conserved quantity $I$ (the temperature for SST) or material via *bulk motion*, *i.e.* for small variations $\Delta x$, $\Delta t$, conservation is expressed as:

$$I(x,t) = I(x + \Delta x, t + \Delta t) \tag{1}$$

applying a first order approximation of the right hand side and moving the resulting terms to the left hand side of equation 1, we obtain the *advection equation*, known also as the *Brightness Constancy Constraint Equation* (BCCE):

$$\frac{\partial I}{\partial t} + (w.\nabla)I = 0 \tag{2}$$

where $\nabla$ denotes the gradient operator, and $w$ the motion vector $\frac{\Delta x}{\Delta t}$. This equation describes the temporal evolution of quantity $I$ for displacement $w$. Note that this equation is also the basis for many variational methods for *Optical Flow*. To retrieve the motion, numerical schemes are applied, and the resulting system of equations, along with a an additional constraint on $w$ is solved for $w$. This motion can then be used to forecast the future value of $I$.

$$\frac{\partial I}{\partial t} + (w.\nabla)I = D\nabla^2 I \tag{3}$$

$\nabla^2$ denotes the Laplacian operator and $D$ the diffusion coefficient. Note that when $D \to 0$, we recover the advection equation 2.

This equation describes a large family of physical processes (e.g. fluid dynamics, heat conduction, wind dynamics, etc). Let us now state a result, characterizing the general solutions of equation 3.

**Theorem 1.** [1] *For any initial condition $I_0 \in L^1(\mathbb{R}^2)$ with $I_0(\pm\infty) = 0$, there exists a unique global solution $I(x, t)$ to the advection-diffusion equation 3:*

$$I(x, t) = \int_{\mathbb{R}^2} k(\,x - w,\, y)\, I_0(y)\, dy \tag{4}$$

*where $k(u, v) = \frac{1}{4\pi Dt} e^{-\frac{1}{4Dt}\|u-v\|^2}$ is a radial basis function kernel, or alternatively, a 2 dimensional Gaussian probability density with mean $u$ and variance $2Dt$.*

Equation 4 provides a principled way to calculate $I(x, t)$ for any time $t$ using the initial condition $I_0$, provided the motion $w$ and the diffusion coefficient $D$ are known. It states that quantity $I(x, t)$ can be computed from the initial condition $I_0$ via a convolution with a Gaussian probability density function. In other words, if $I$ was used as a model for the evolution of the SST and the surface's underlying advecting mechanisms were known, future surface temperatures could be predicted from previous ones. Unfortunately neither the initial conditions, the motion vector nor the diffusion coefficient are known. They have to be estimated from the data. Inspired from the general form of solution 4, we propose a ML method, expressed as a Deep Learning architecture for predicting SST. This model will learn to predict a motion field analog to the $w$ in equation 4, which will be used to predict future images.

## 3  MODEL

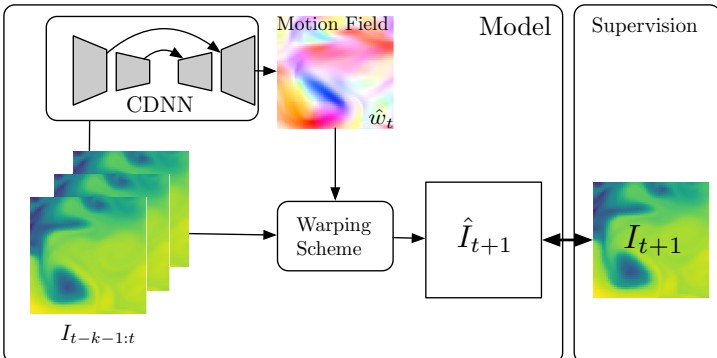

Figure 1: Motion is estimated from the input images ($I_{t-k-1:t}$) with a convolutional neural network (top left CDNN component). A warping scheme then displaces the last input image along this motion estimate to produce the future image. The error signal is calculated using the target future image $I_{t+1}$, and is backprogated through the warping scheme to correct the CDNN. To produce multiple time-step forecasts, the predicted image is fed back in the CDNN in an autoregressive manner.

The model consists of two main components, as illustrated in Figure 1. One predicts the motion field from a sequence of past input images, this is convolutional-deconvolutional (CDNN) module on the top of figure 1, and the other warps the last input image using the motion field from the first component, in order to produce an image forecast. The entire system is trained in an end-to-end fashion, using only the supervision from the target SST image. By doing so, we are able to produce an interpretable latent state which corresponds in our problem to the velocity field advecting the temperatures.

Let us first introduce some notations. Each SST image $I_t$ is acquired on a bounded rectangle of $\mathbb{R}^2$, named $\Omega$. We denote $I_t(x)$ and $w_t(x)$ the sea surface temperature and the two-dimensional motion vector at time $t \in \mathbb{R}$ at position $x \in \Omega$. $I_t : \Omega \to \mathbb{R}$ and $w_t : \Omega \to \mathbb{R}^2$ represent the temperatures and the motion vector field at time $t$ defined on $\Omega$. When time $t$ and position $x$ are available from the context, we will drop the subscript $t$ from $w_t(x)$ and $I_t(x)$, along with $x$ for clarity. Given a sequence of $k$ consecutive SST images $\{I_{t-k-1}, ..., I_t\}$ (also denoted as $I_{t-k-1:t}$), our goal is to predict the next image $I_{t+1}$.

---

[1]A proof of the theorem is provided in appendix A

## 3.1 MOTION ESTIMATION

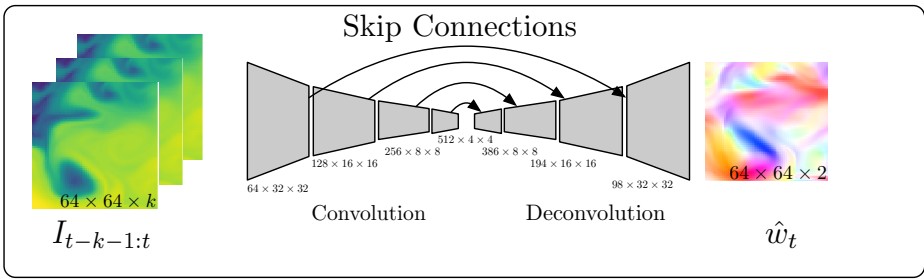

Figure 2: Architecture of the CDNN motion estimation component. For the estimated motion flow $\hat{w}_t$, colours correspond to the flow orientation and colour intensity to the flow intensity

As indicated in section 2, provided the underlying motion field is known, one can compute SST forecasts. Let us introduce how the motion field is estimated in our architecture. We are looking for a vector field $w$ which when applied to the geometric space $\Omega$ renders $I_t$ close to $I_{t+1}$, i.e. $I_{t+1}(x) \simeq I_t(x + w(x))$, $\forall x \in \Omega$. If $I_{t+1}$ were known, we could estimate $w$, but $I_{t+1}$ is precisely what we are looking for. Instead, we choose to use a convolutional-deconvolutional architecture to predict a motion vector for each pixel. As shown in figure 2, this network makes use of skip connections He et al. (2015), allowing fine grained information from the first layers to flow through in a more direct manner. We use a *Batch Normalization* layer between each convolution, and *Leaky ReLU* (with parameter value set to $0.1$) non-linearities between convolutions and transposed-convolutions. We used $k = 4$ concatenated images $I_{t-k-1:t}$ as input for training. We have selected this architecture experimentally, testing different state-of-the-art convolution-deconvolution network architectures. Let $\hat{w} \in \mathbb{R}^{2 \times W \times H}$ be the output of the network, where $W$ and $H$ are respectively the width and height of the images, and '2' corresponds to the two components of the flow at each point of the motion field.

Generally, and this is the case for our problem, we do not have a direct supervision on the motion vector field, since the target motion is usually not available. Using the warping scheme introduced below, we will nonetheless be able to (weakly) supervise $w$, based on the discrepancy of the warped version of the $I_t$ image and the target image $I_{t+1}$.

## 3.2 WARPING SCHEME

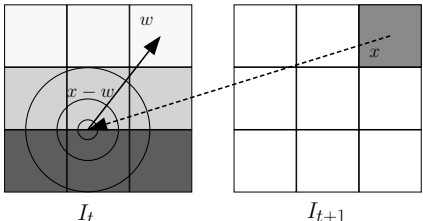

Figure 3: Warping scheme. To calculate the pixel value for time $t+1$ at position $x$, we first compute its previous position at time $t$, *i.e.* $x - w$. We then center a Gaussian in that position in order to obtain a weight value for each pixel in $I_t$ based on its distance with $x - w$, and compute a weighted average of the pixel values of $I_t$. This weighted average will correspond to the new pixel value at $x$ in $I_{t+1}$.

Discretizing the solution of the advection-diffusion equation in section 2 by replacing the integral with a sum, and setting image $I_t$ as the initial condition, we obtain a method to calculate the future image, based on the motion field estimate $\hat{w}$. The latter is used as a warping scheme:

$$\hat{I}_{t+1}(x) = \sum_{y \in \Omega} k(\, x - \hat{w}(x),\, y)\, I_t(y) \tag{5}$$

where $k(x - \hat{w}, y) = \frac{1}{4\pi D \Delta t} e^{-\frac{1}{4D\Delta t}\|x - \hat{w} - y\|^2}$ is a radial basis function kernel, as in equation 4, parameterized by the diffusion coefficient $D$ and the time step value $\Delta t$ between $t$ and $t+1$ and $\hat{w}$ is the estimated value of the vector flow $w$. To calculate the temperature for time $t + 1$ at position $x$, we compute the scalar product between $k(x - \hat{w}, .)$, a Gaussian centered in $x - \hat{w}$, and the previous image $I_t$. Simply put, it is a weighted average of the temperatures $I_t$, where the weight values are larger when the pixel's positions that are closer to $x - \hat{w}$. Informally, $x - \hat{w}$ corresponds to the pixel's previous position at time $t$. See figure 3.

As seen by the relation with the solution of the advection-diffusion equation, the proposed warping mechanism is then clearly adapted to the modeling of phenomena governed by the advection-diffusion equation. SST forecasting is a particular case, but the proposed scheme can be used for any problems in which advection and diffusion are occurring. Moreover, this warping scheme is entirely differentiable, allowing backpropagation of the error signal to the motion fireld estimating module.

This warping mechanism has been inspired by the Spatial Transformer Network (STN) Jaderberg et al. (2015), originally designed to be incorporated as a layer in a convolutional neural network architecture in order to gain invariance under geometric transformations. Using the notations in Jaderberg et al. (2015), when the inverse geometric transformation $\mathcal{T}_\theta$ of the grid generator step is set to $\mathcal{T}_\theta(x) = x - \hat{w}(x)$, and the kernels $k(\,.\,; \Phi_x)$ and $k(\,.\,; \Phi_y)$ in the sampling step are radial basis function kernels, we recover our warping scheme. The latter can be seen as a specific case of the STN, without the localization step. This result theoretically grounds the use of the STN for Optical Flow in many recent articles Zhu et al. (2017), Yu et al. (2016), Patraucean et al. (2015), Finn et al. (2016): in equation 3, when $D \to 0$, we recover the brightness constancy constraint equation, used in the latter.

For training, supervision is provided at the output of the warping module. It consists in minimizing the discrepancy between the warped image $\hat{I}_{t+1}$ and the target image $I_{t+1}$. The loss is measured via a differentiable function and the gradient is back propagated through the warping function in order to adjust the parameters of the convolutional-deconvolutional module generating the vector field. This is detailed in the next section.

### 3.3 Loss function

At each iteration, the model aims at forecasting the next observation, given the previous ones. We evaluate the discrepancy between the warped image $\hat{I}_{t+1}$ and the target image $I_{t+1}$ using the Charbonnier penalty function $\rho(x) = (x + \epsilon)^{\frac{1}{\alpha}}$, where $\epsilon$ and $\alpha$ are parameters to be set. Note that with $\epsilon = 0$ and $\alpha = \frac{1}{2}$, we recover the $\ell_2$ loss. The Charbonnier penalty function is known to reduce the influence of outliers compared to an $l_2$ norm. We have also tested the Laplacian pyramid loss Ling & Okada (2006), where we enforce convolutions of all deconvolutional layers to be close to down-sampled versions of the target image in the Charbonnier penalty sense, but we have observed an overall decrease in generalization performance.

The proposed NN model has been designed according to the intuition gained from general background knowledge of a physical phenomenon, here advection-diffusion equations. Additional prior knowledge – expressed as partial differential equations, or through constraints – can be easily incorporated in our model, by adding penalty terms in the loss function. As the displacement $w$ is explicitly part of our model, one strength of our model is its capacity to apply some regularization

term directly on the motion field. In our experiments, we tested the influence of different terms: divergence $\nabla. w_t(x)^2$, magnitude $\|w_t(x)\|^2$ and smoothness $\|\nabla w_t(x)\|^2$.

$$L_t = \sum_{x \in \Omega} \rho(\hat{I}_{t+1}(x) - I_{t+1}(x)) + \lambda_{\mathrm{div}}(\nabla. w_t(x))^2 + \lambda_{\mathrm{magn}} \|w_t(x)\|^2 + \lambda_{\mathrm{grad}} \|\nabla w_t(x)\|^2 \quad (6)$$

## 4 EXPERIMENTS

### 4.1 DATASET DESCRIPTION

Since 1982, high resolution SST data has been made available by the *NOAA6* weather satellite, Bernstein (1982). Dealing directly with these data requires a lot of preprocessing (e.g. some regions are not available due to clouds, hindering temperature acquisition). In order to avoid such complications which are beyond the scope of this work, we used synthetic but realistic SST data of the Atlantic ocean generated by a sophisticated simulation engine: NEMO (Nucleus for European Modeling of the Ocean) engine [2], Madec (2008). NEMO is a state-of-the-art modelling framework of ocean related engines. It is a primitive equation model adapted to the regional and global ocean circulation problems. Historical data is accumulated in the model to generate a synthesized estimate of the states of the system using data *analysis*, a specific data assimilation scheme, which means that the data does follow the true temperatures. The resulting dataset is constituted of daily temperature acquisitions of 481 by 781 pixels, from 2006-12-28 to 2017-04-05 (3734 acquisitions).

We extract 64 by 64 pixel sized sub-regions as indicated in figure 4.1. We use data from years 2006 to 2015 for training and validation (94743 training examples), and years 2016 to 2017 for testing. We withhold 20% of the training data for validation, selected uniformly at random at the beginning of each experiment. For the tests we used sub-regions enumerated 17 to 20 in figure 4.1, where the interactions between hot and cold waters make the dynamics interesting to study. All the regions numbered in figure 4.1, from 2006 to 2015 where used for training [3]. Each sequence of images used for training or for evaluation corresponds to a specific numbered sub-region. We make the simplifying hypothesis that the data in a single sub-region contains enough information to forecast the future of the sub-region. As the forecast is for a small temporal horizon we can assume that the influence from outside the region is small enough.

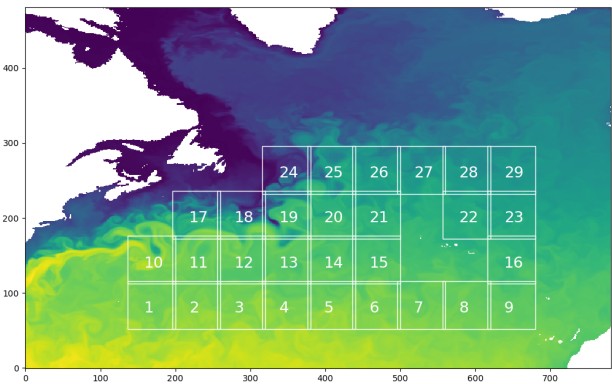

Figure 4: Sub regions extracted for the dataset. Test regions are regions 17 to 20.

We normalize the daily SST acquisitions of each sub region using the mean and the standard deviation of all the SST data of the sub-region acquired on the same day of the year for all the years in the

---

[2]NEMO data are available at `http://marine.copernicus.eu/services-portfolio/ access-to-products/?option=com_csw&view=details&product_id=GLOBAL_ ANALYSIS_FORECAST_PHY_001_024`

[3]non numbered regions correspond to land and not sea on the figure

training set, *i.e.* the SST acquisition of sub-region 2 on date September 8th 2017 is standardized using the data of all the September 8th available in the dataset. This removes the seasonal component from SST data.

## 4.2 BASELINE COMPARISON

We compare our model with several baselines. Each model is evaluated with a mean square error metric, forecasting images on a horizon of 6 (we forecast from $I_{t+1}$ to $I_{t+6}$ and then average the MSE). The hyperparameters are tuned using the validation set. Neural network based models are run on a Titan Xp GPU, and runtime is given for comparison purpose.

Concerning the constraints on the vector field $w$ (equation 6. the regularization coefficients selected via validation are $\lambda_{\text{div}} = 1$, $\lambda_{\text{magn}} = -0.03$ and $\lambda_{\text{grad}} = 0.4$. The coefficient diffusion $D$ was set to 0.45 by cross validation. We also compare the results with the model without any regularization.

Our reference model for forecasting is Béréziat & Herlin (2015), a numerical assimilation model which relies on data assimilation. In Béréziat & Herlin (2015), the ocean's dynamics are modeled using shallow water equations Vallis (2017) and the initial conditions, along with other terms, are estimated using assimilation techniques Trémolet (2006). This is a state of the art assimilation model for predicting ocean dynamics, here SST.

The other baselines are 1) an autoregressive convolutional-deconvolutional NN (ACNN), with an architecture similar to our CDNN module, but trained to predict the future image directly, without explicitly representing the motion vector field. Each past observation is used as an input channel (the 4 input images used in the experiments are concatenated), and the output is used as new input for multi step forecasting, 2) a ConvLSTM model Shi et al. (2015), which uses convolutional transitions in the inner LSTM module, and 3) the model in Mathieu et al. (2015) which is a multi-scale ACNN trained as a Generative Adversial Network (GAN). We have used a non-official code for Mathieu et al. (2015), which is made available at `https://github.com/dyelax/Adversarial_Video_Generation`. For Béréziat & Herlin (2015), the code has been provided by the authors of the paper. We have implemented the ACNN and ConvLSTM models ourselves. The code for our models, along with these baselines will be made available.

## 4.3 QUANTITATIVE RESULTS

| Model | Average Score (MSE) | Average Time |
|---|---|---|
| Numerical model Béréziat & Herlin (2015) | 1.99 | 4.8 s |
| ConvLSTM Shi et al. (2015) | 5.76 | 0.018 s |
| ACNN | 15.84 | 0.54 s |
| GAN Video Generation (Mathieu et al. (2015)) | 4.73 | 0.096 s |
| Proposed model with regularization | **1.42** | 0.040 s |
| Proposed model without regularization | 2.01 | 0.040 s |

Table 1: Average score and average time on test data. Average score is calculated using the *mean square error* metric (MSE), time is in seconds. The regularization coefficients for our model have been set using a validation set with $\lambda_{\text{div}} = 1$, $\lambda_{\text{magn}} = -0.03$ and $\lambda_{\text{grad}} = 0.4$.

Quantitatively, our model performs well. The MSE score is better than any of the baselines. The closest NN baseline is Mathieu et al. (2015) which regularizes a regression convolution-deconvolution model with a GAN. The performance is however clearly below the proposed model and it does not allow to easily incorporate prior constraints inspired from the physics of the phenomenon. ACNN is a direct predictor of the image sequence, implemented via a CDNN module identical to the one used in our model. Its performance is poor. Clearly, a straightforward use of prediction models is not adapted to the complexity of the phenomenon. ConvLSTM performs better: as opposed to the ACNN, it seems to be able to capture a dynamic, although not very accurately. Overall, direct prediction models are not able to capture the complex underlying dynamics and they produce blurry sequences of images. The GAN explicitly forces the network output to eliminate the blurring effect and then makes it able to capture short term dynamics. The state of the art numerical

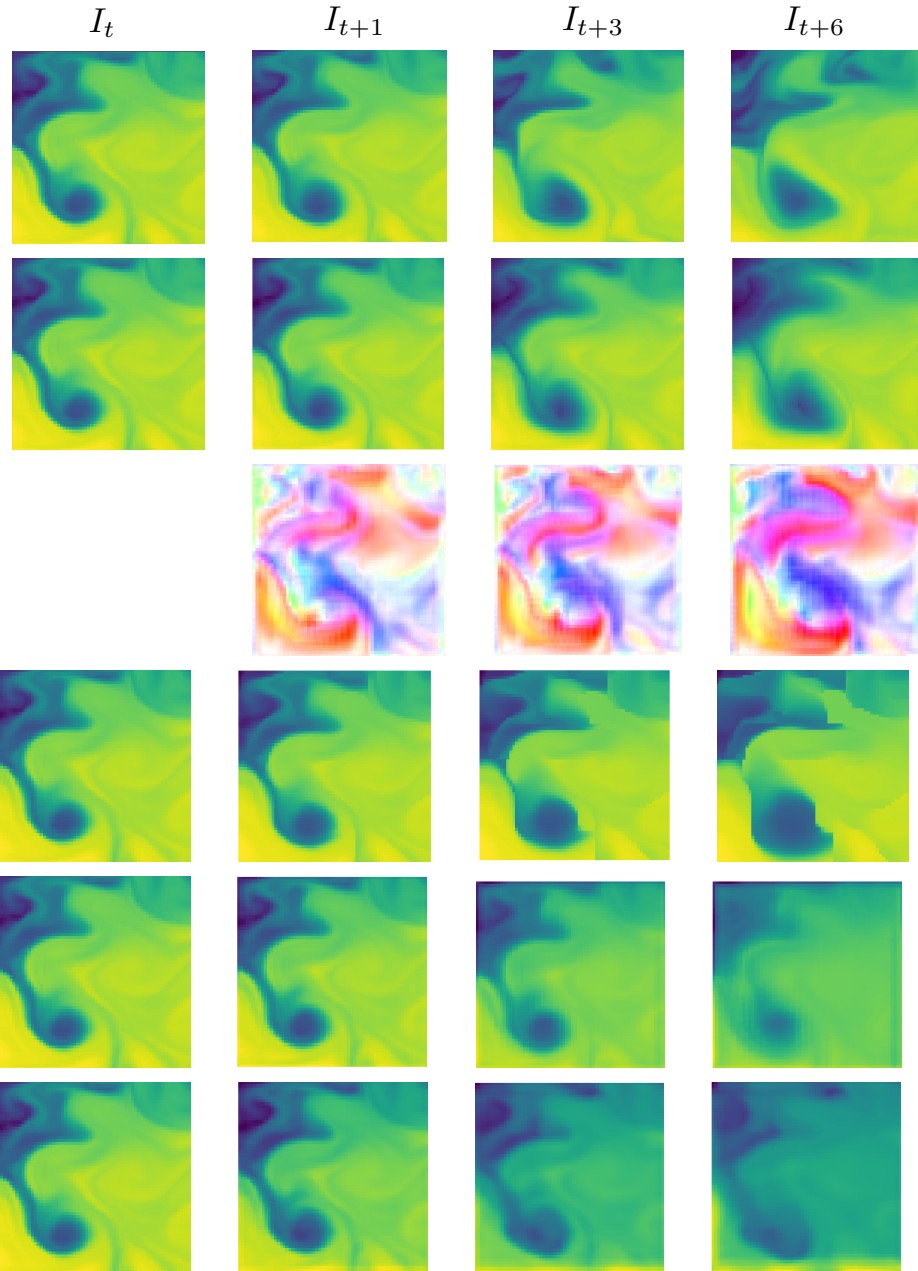

Figure 5: From top to bottom: target, our model prediction, our model flow, numerical assimilation model , ACNN, *ConvLSTM*. Data correspond to daily temperatures from January 17 to January 23, 2017

model Béréziat & Herlin (2015), performs well but has a slightly lower performance than our regularized model, although it incorporates more prior constraints. This shows that pure ML models, when conceived adequately and when trained with enough data, can be competitive with state of the art dedicated models. Regularizing the motion vector $w$ notably increases the performance w.r.t. to the unregularized model. The choice of the constraints (divergence, magnitude and smoothness) inspired here by physical background correspond to relevant priors on the dynamics of the model.

As for the running time, the proposed model is extremely fast, being just above the ConvLSTM model of Shi et al. (2015). The running time of Béréziat & Herlin (2015)'s model is not comparable to the others. It was run on a CPU (no GPU code) when all the others were run on Titan Xp GPU. However, an optimization procedure is required to estimate the motion field, and it is clearly slower than the straightforward NN predictions. Moreover, in order to prevent the numerical scheme from diverging, multiple intermediate forecasts are required.

Besides *MSE*, we need to analyze the prediction samples qualitatively. Figure 4.3 shows the predictions obtained by the different models. On the top row, the ground truth for a sequence of 4 temperature images corresponding to time $t$, $t + 1$, $t + 3$ and $t + 6$. The second row corresponds to our regularized model prediction at times $t + 1$, $t + 3$ and $t + 6$ (time $t$ corresponds to the last input image, it is repeated on each row). The model seems to conserve temperatures. The prediction is close to the target for $t + 1$, $t + 3$ and starts to move away at time $t + 6$. The third row shows the motion flow estimated by the model. Each color in the flow images corresponds to a motion vector. There is clearly a strong evolving dynamic captured for this sequence. Row 4 is the numerical assimilation model of Béréziat & Herlin (2015). It also clearly captures some dynamics and shows interesting patterns, but it tends to diverge when the prediction horizon increases. The ACNN model (row 5) rapidly produces blurry images; it does not preserve the temperatures and does not seem to capture any dynamics. On row 6 are plotted the predictions of the ConvLSTM model. Temperature is not preserved and although a dynamic is captured, it does not correspond to the target. Overall, the proposed model seems to forecast SST quite accurately, while retrieving a coherent motion vector field.

# 5 RELATED WORK

**ML for Physical modeling** Close to this work is the field of spatio-temporal statistics. In their reference book Cressie & Wikle (2015) also advocate the use of physical background knowledge to build statistical models. They show how the design of statistical models can be inspired from partial differential equations linked to an observed physical phenomenon. They mainly consider auto-regressive models within a hierarchical Bayesian framework. In Raissi et al. (2017), Archambeau et al. (2007) and Alvarez et al. (2011) the author use PDE-inspired gaussian process to model physical process. Even if the methods and the application are different, the motivation and arguments are similar to the ones developed here.

Another interesting research direction is the use of NNs for reducing the complexity of numerical simulation for physical processes. Generally, in these approaches statistical models are used in place of a computational demanding component of the numerical simulation process. For example in the domain of fluid dynamics, Tompson et al. (2017) and Ladický et al. (2015) propose to use regressors for simulating fluid and smoke animation. Ladický et al. (2015) use a random forest to compute particle location and Tompson et al. (2017) use a CNN to approximate part of a numerical PDE scheme. In these approaches, ML is only a component of a numerical simulation scheme whereas we aim at modeling the whole physical process via a Deep Learning approach. Farther to our objective, Rudy et al. (2017) make use of a sparse regression method for discovering the governing partial differential equation(s) of a given system by time series measurements in the spatial domain. Other works have suggested using neural networks for physical process forecast, such as Brajard et al. (2017).

Our work is also related to recent developments in computer vision, in the related but distinct fields of video prediction and motion estimation in videos. Our goal and the domain of application are clearly different from video modeling, but since our solution involves predicting a motion field and the next SST image, the solutions share some similarities. Motion estimation and video predictions by deep architectures have motivated a series of work over the last two years. We briefly review them below and outline the differences.

**Optical Flow** Optical flow consists in retrieving the apparent motion of objects, surfaces, or particles between two consecutive frames of a video. The extracted motion can be used in many areas such as object detection, object tracking, movement detection, robot navigation and visual odometry. In the vision community, this is considered as a problem by itself and several papers are dedicated to this topic. Classical methods rely on the brightness constancy constrain equation (BCCE) (equation 2), derived from the observation that surfaces usually persist over time and hence the intensity value of a small region remains the same despite its position change Sun et al. (2008). Since using BCCE directly leads to complicated optimizing issues, classic approaches – namely differential methods – approximate BCCE using a first order Taylor expansion and develop variational methods.

As an alternative to these methods, Deep Learning models have been recently proposed for estimating the optical flow between 2 images. Fischer et al. (2015) formulate optical flow as a supervised regression problem, using a CNN to predict motion. Ilg et al. (2016) build on this approach and propose to use an ensemble of these CNN architectures. They assess results on par with state of the art methods for optical flow, while maintaining a small computational overhead. The difficulty here is that these methods require a notable quantity of target data, i.e. optical flow images, while because of the complexity of manually annotating flow images, there are only a few small annotated collections available. Fischer et al. (2015) and Ilg et al. (2016) chose to pretrain their model on a synthetic dataset made of computer animations and their associated motion and show that this information transfers well to real videos. Yu et al. (2016) demonstrate that it is possible to predict the optical flow between two input images in an unsupervised way using a CNN and a Spatial Transformer Network. This is however not extensible for prediction as is done in our setting since this requires the two images $I_t$ and $I_{t+1}$ as input while $I_{t+1}$ is not available at inference time for prediction.

**Video prediction**

It is only very recently that video prediction emerged as a task in the Deep Learning community. For this task, people are generally interested at predicting accurately the displacement/ emergence/ disappearing of objects in the video. In our application, the goal is clearly different since we are interested into modeling the whole dynamics behind image changes and not at following moving objects. Let us first introduce some methods that perform prediction by computing optical flow or a similar transformation. Both Patraucean et al. (2015) and Finn et al. (2016) use some form of motion flow estimation. For next frame prediction Patraucean et al. (2015) introduce a STN module at the hidden layer of a LSTM in order do estimate a motion field in this latent space. The resulting image is then decoded in the original image space for prediction. This approach clearly does not allow introducing prior knowledge on the field vector as this has been done in our work. Finn et al. (2016) learn affine transformations on image parts in order to predict object displacement and Van Amersfoort et al. (2017) proposed a similar model.

Let us now consider models that directly attempt to predict the next frame without estimating a motion field. As shown in the experimental section, plain application of autoregressive models produces blurred images. Mathieu et al. (2015), one of our baseline proposed to use different loss functions and a GAN regularization of a CDNN predictor which led to sharper and higher quality predictions. Significant improvements have been obtained with the Video Pixel Network of Kalchbrenner et al. (2016), which is a sophisticated architecture composed of resolution preserving CNN encoders, LSTM and PixelCNN decoders which form a conditional Spatio-temporal video autoencoder with differentiable memory. This model is probably state of the art today for video prediction, They reach a high accuracy on moving MNIST and good performance on a robot video dataset. A drawback is the complexity of the model and the number of parameters: they are using respectively 20 M and 1 M frames on these two datasets. We did not test this model since up to our knowledge no code was available.

## 6 CONCLUSION AND FUTURE WORK

The evolution in time of the proposed model is deterministic. Predicting future observations should also deal with the inherent ambiguity and lack of information for the prediction task. A natural future direction would be to incorporate uncertainty in the model's evolution in the proposed framework. We can extend the proposed model by incorporating a stochastic latent variable in the flow field generation process. A promising direction is the development of generative models which has become popular in Deep Learning, leading to different families of innovative models. For example,

the Stochastic Gradient Variational Bayes algorithm (SGVB) Kingma & Welling (2014) provides a framework for learning stochastic latent variables with deep neural networks, and has recently been used by some authors to model time series Karl et al. (2016); Chung et al. (2015); Krishnan et al. (2015). A recent work where both spatial and temporal information are considered is Walker et al. (2016) who model pixel trajectories in a video. As a follow up of our work, we plan to consider such extensions in the future.

The data intensive paradigm offers alternative directions to the classical physical approaches for modeling complex natural processes. Our belief is that cross fertilization of both paradigms is essential for pushing further the frontier of complex data modeling. By using as an example application a relatively complex problem concerning ocean dynamics, we proposed a principled way to design Deep Learning models using inspiration from the physics. The proposed approach can be easily generalized to a class of problems for which the underlying dynamics follow advection-diffusion principles. We have compared the proposed approach to a series of baselines. It is able to reach performance comparable to a state of the art numerical model and clearly outperform alternative NN models used as baselines.

ACKNOWLEDGMENTS

This work was partially funded by ANR project LOCUST - ANR-15-CE23-0027 and by CLEAR - Center for LEArning & data Retrieval - joint lab. With Thales (www.thalesgroup.com).

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

## A   PROOF OF THE THEOREM IN SECTION 2 1

*Proof.* In the following, bold $\mathbf{x}$ and $\mathbf{y}$ will denote vectors of $\mathbb{R}^2$, while $x$ and $y$ will correspond to the first and second components of $\mathbf{x}$, respectively. Analogously, $u$ and $v$ will correspond to the components of $w$. The 2D Fourier Transformation $\mathcal{F}$ of $f : \mathbb{R}^2 \to \mathbb{R}$ is defined as

$$
\begin{aligned}
\mathcal{F}(f) &= \int_{\mathbb{R}^2} f(\mathbf{x}) e^{-i<\xi,\mathbf{x}>} d\mathbf{x} \\
&= \int_{\mathbb{R}} \int_{\mathbb{R}} f(x,y) e^{-ix\xi_1 - iy\xi_2} dx dy
\end{aligned}
\tag{7}
$$

We apply the Fourier Transform $\mathcal{F}$ to both sides of 3. As consequence of the linearity of the Fourier transform, we can calculate decompose the Fourier transform of the left hand side in the sum of the transforms of each term. We have three terms: $\frac{\partial I}{\partial t}$, $(w.\nabla)I$ and $-D\nabla^2 I$.

$$
\begin{aligned}
\mathcal{F}(\frac{\partial I}{\partial t}) &= \int_{\mathbb{R}^2} \frac{\partial I}{\partial t} e^{-i<\mathbf{x},\xi>} d\mathbf{x} \\
&= \int_{\mathbb{R}^2} \frac{\partial}{\partial t} (I e^{-i<\mathbf{x},\xi>}) d\mathbf{x} \\
&= \frac{\partial}{\partial t} \int_{\mathbb{R}^2} I e^{-i<\mathbf{x},\xi>} d\mathbf{x} \\
&= \frac{\partial \mathcal{F}(I)}{\partial t}
\end{aligned}
\tag{8}
$$

$$
\begin{aligned}
\mathcal{F}((w.\nabla)I) &= \int_{\mathbb{R}^2} (w.\nabla)I e^{-i<\mathbf{x},\xi>} d\mathbf{x} \\
&= \int_{\mathbb{R}} \int_{\mathbb{R}} (u\frac{\partial I}{\partial x} + v\frac{\partial I}{\partial y}) e^{-ix\xi_1 - iy\xi_2} dx dy \\
&= u \int_{\mathbb{R}} e^{-iy\xi_2} \int_{\mathbb{R}} \frac{\partial I}{\partial x} e^{-ix\xi_1} dx dy + v \int_{\mathbb{R}} e^{-ix\xi_1} \int_{\mathbb{R}} \frac{\partial I}{\partial y} e^{-iy\xi_2} dy dx \\
&= i\xi_1 u \int_{\mathbb{R}} e^{-iy\xi_2} \int_{\mathbb{R}} I e^{-ix\xi_1} dx dy + i\xi_2 v \int_{\mathbb{R}} e^{-ix\xi_1} \int_{\mathbb{R}} \frac{\partial I}{\partial y} e^{-iy\xi_2} dy dx \\
&= i\xi_1 u \int_{\mathbb{R}} \int_{\mathbb{R}} I e^{-ix\xi_1 - iy\xi_2} dx dy + i\xi_2 v \int_{\mathbb{R}} \int_{\mathbb{R}} I e^{-ix\xi_1 - iy\xi_2} dx dy \\
&= (i\xi_1 u + i\xi_2 v) \int_{\mathbb{R}} \int_{\mathbb{R}} I e^{-ix\xi_1 - iy\xi_2} dx dy \\
&= i<\xi, w> \mathcal{F}(I)
\end{aligned}
\tag{9}
$$

$$
\begin{aligned}
\mathcal{F}(-D\nabla^2 I) &= -\int_{\mathbb{R}^2} D\nabla^2 I e^{-i<\mathbf{x},\xi>} d\mathbf{x} \\
&= -\int_{\mathbb{R}} \int_{\mathbb{R}} D(\frac{\partial^2 I}{\partial x^2} + \frac{\partial^2 I}{\partial y^2}) e^{-ix\xi_1 - iy\xi_2} dx dy \\
&= -D \int_{\mathbb{R}} e^{-iy\xi_2} \int_{\mathbb{R}} \frac{\partial^2 I}{\partial x^2} e^{-ix\xi_1} dx dy - D \int_{\mathbb{R}} e^{-ix\xi_1} \int_{\mathbb{R}} \frac{\partial^2 I}{\partial y^2} e^{-iy\xi_2} dy dx \\
&= -(i\xi_1)^2 D \int_{\mathbb{R}} e^{-iy\xi_2} \int_{\mathbb{R}} I e^{-ix\xi_1} dx dy - (i\xi_2)^2 D \int_{\mathbb{R}} e^{-ix\xi_1} \int_{\mathbb{R}} I e^{-iy\xi_2} dy dx \\
&= D\xi_1^2 \int_{\mathbb{R}} e^{-iy\xi_2} \int_{\mathbb{R}} I e^{-ix\xi_1} dx dy + D\xi_2^2 \int_{\mathbb{R}} e^{-ix\xi_1} \int_{\mathbb{R}} I e^{-iy\xi_2} dy dx \\
&= D\xi_1^2 \int_{\mathbb{R}} \int_{\mathbb{R}} I e^{-ix\xi_1 - iy\xi_2} dx dy + D\xi_2^2 \int_{\mathbb{R}} \int_{\mathbb{R}} I e^{-ix\xi_1 - iy\xi_2} dx dy \\
&= D \|\xi\|^2 \mathcal{F}(I)
\end{aligned}
\tag{10}
$$

Regrouping all three previously calculated terms, we obtain

$$\frac{\partial \mathcal{F}(I)}{\partial t} + (i <\xi, w> + D \|\xi\|^2)\mathcal{F}(I) = 0 \tag{11}$$

This is a first order ordinary differential equation of the form $f'(t) + af(t) = 0$, which admits a known solution $f(t) = f(0)e^{-at}$. Thus, the solution of 11 is

$$\begin{aligned}
\mathcal{F}(I) &= \mathcal{F}(I)_0 \, e^{-(i<\xi,w>+D\|\xi\|^2)t} \\
&= \mathcal{F}(I)_0 \, e^{-i<\xi,w>t} e^{-Dt\|\xi\|^2}
\end{aligned} \tag{12}$$

where $\mathcal{F}(I)_0$ denotes the initial condition of the advection diffusion equation in the frequency domain. In order to obtain a solution of 3 in the spatial domain, we calculate the inverse Fourier Transform $\mathcal{F}^{-1}$ of 12. The multiplication of two functions in the frequency domain is equivalent to their convolution in the spatial domain, i.e. $\mathcal{F}(f * g) = \mathcal{F}(f)\mathcal{F}(g)$. Hence, the inverse of both terms $\mathcal{F}(I)_0 \, e^{-i<\xi,w>t}$ and $e^{-Dt\|\xi\|^2}$ can be calculated separately:

Multiplication by a complex exponential in the frequency domain is equivalent to a shift in the spatial domain : $e^{-i<\xi,w>}\mathcal{F}(f(\mathbf{x})) = \mathcal{F}(f(\mathbf{x} - w))$, for $v \in \mathbb{R}^2$. Thus, for the first term,

$$\mathcal{F}^{-1}(\mathcal{F}(I)_0 \, e^{-(i<\xi,w>)t}) = I_0(\mathbf{x} - w) \tag{13}$$

For the second term, we use the fact that the Fourier Transform of a Gaussian function also is a Gaussian function, i.e. $\mathcal{F}(\frac{1}{2\pi\sigma^2}e^{-\frac{1}{2\sigma^2}\|\mathbf{x}\|^2}) = e^{-\frac{1}{2}\sigma^2\|\xi\|^2}$. Identifying $\sigma^2$ with $2Dt$, we have:

$$\mathcal{F}^{-1}(e^{-Dt\|\xi\|^2}) = \frac{1}{4\pi Dt}e^{-\frac{1}{4Dt}\|\mathbf{x}\|^2} \tag{14}$$

As has been stated above, the solution is a convolution of both previously calculated terms:

$$\begin{aligned}
I(\mathbf{x}, t) &= \int_{\mathbb{R}^2} \frac{1}{4\pi Dt}e^{-\frac{1}{4Dt}\|\mathbf{y}\|^2} I_0(\mathbf{x} - w - \mathbf{y})dy \\
&= \int_{\mathbb{R}^2} \frac{1}{4\pi Dt}e^{-\frac{1}{4Dt}\|\mathbf{x}-w-\mathbf{y}\|^2} I_0(\mathbf{y})d\mathbf{y}
\end{aligned} \tag{15}$$

$\square$

## B  On the Generalization in Space and Time

The ability of the model to adapt to other conditions should be evaluated on other regions. This, however, requires a complete study by itself and is beyond the scope of this paper. We, however, present below complementary experiments aimed at assessing the potential of the proposed model for forecasting SST on sequences distant in time and space from the ones used for training.

### B.1  Temporal Dimension

In section 4, training has been performed on data from 2006 - 2015 and testing on the period 2016-2017. In order to provide some indication of the model behavior on more distant time intervals between train and test data, we have performed experiments using the same regions (17 to 20) as in section 4, but using the period 2011 to 2017 for training and period 2006 to 2010 for testing. Figure B.1 shows the MSE curve on this test set, each point corresponding to the mean MSE on predictions performed on 6 days ahead the current date. The most important conclusion is probably that the MSE error remains in the same range for all these years. All the yearly error curve show a clear seasonal phenomenon with a higher prediction error during summer. A similar behavior has been observed when exchanging train and test data.

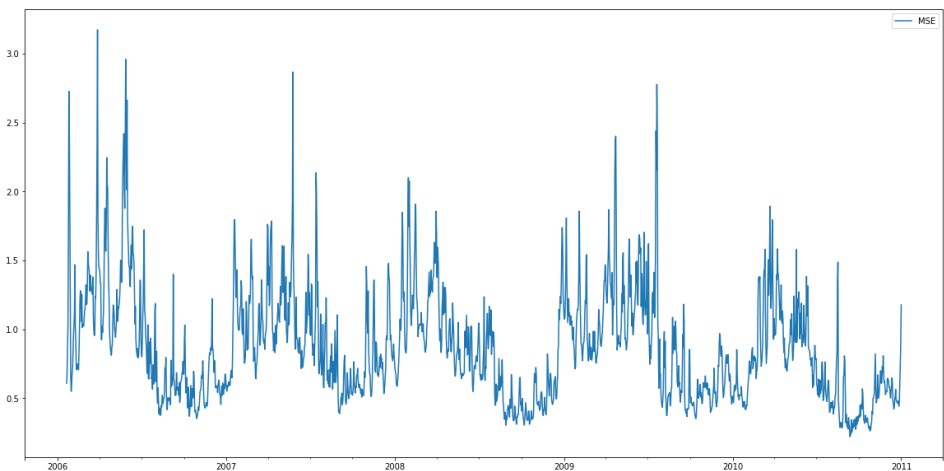

Figure 6: Evaluation of our model's accuracy in time on data from 2006 to 2010 using data from 2011 to 2017 for training. Regions 17 to 20 were used for both periods. Each day, we produce daily forecasts for 6 days ahead and calculate the associated mean square error.

### B.2  Spatial Dimension

In the experiments, the models have been trained and evaluated on selected regions (numbered 17 to 20 in Figure 4.1), considered as the most interesting for the observed dynamics.

|  | Test Regions 17 & 18 | Test Regions 8 & 9 |
|---|---|---|
| Model trained on Regions 17 & 18 | 1.43 | 1.22 |
| Model trained on Regions 8 & 9 | 1.90 | 1.19 |

Table 2: Evaluation of our model's spatial generalization ability. We train our model on two distinct regions and calculate the MSE on both regions for each trained model.

We describe below some results providing indications on how the model performs on regions different from the training ones. For these experiments, the model has been trained on regions 17 and 18 in Figure 4.1 and tested on two other regions (regions 8 and 9), and vice versa (trained on 8 and 9 and tested on 17 and 18). The two couples of regions have been selected so as to have different latitude and longitude. The underlying physical processes generating the data are known to be different in these regions: the overall motion in regions 17 and 18 is greater, and the difference between extreme

temperature is larger, compared to regions 8 and 9. Experimental conditions are similar to the one described in section 4, i.e. 2006-2015 have been used for training and 2016-1017 for testing.

Results in Table B.2 show that the model generalizes reasonably well to unseen data from distant spatial regions, with a slight decrease in performance when training and test regions do not correspond. The performance loss is 0.47 for regions (17, 18) which show a strong dynamics, whereas it is only 0.03 for regions (8, 9) for which the dynamics are more stable. Most notably, MSE performance depends more on the region itself than on the train/ test conditions. Error is always higher in regions with strong dynamics (17, 18) than on more stable regions (8, 9) whatever the train/ test conditions are. Note that to further improve the results on distant data, it is possible to fine-tune the model using data from the studied regions.

## C ADDITIONAL SAMPLES FROM OUR MODEL

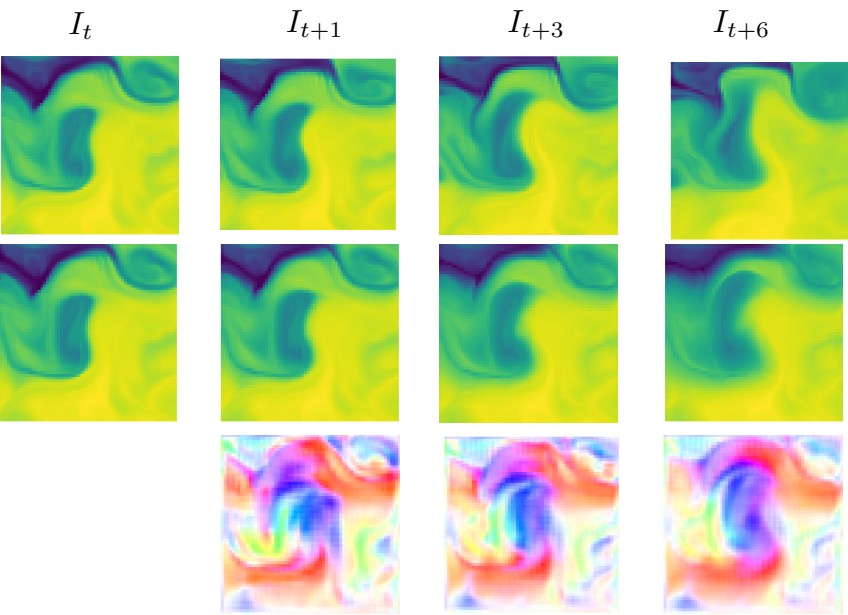

Figure 7: Output for the 6 of May to the 9 of May 2016, Output , From top to bottom: target, our model prediction, our model flow

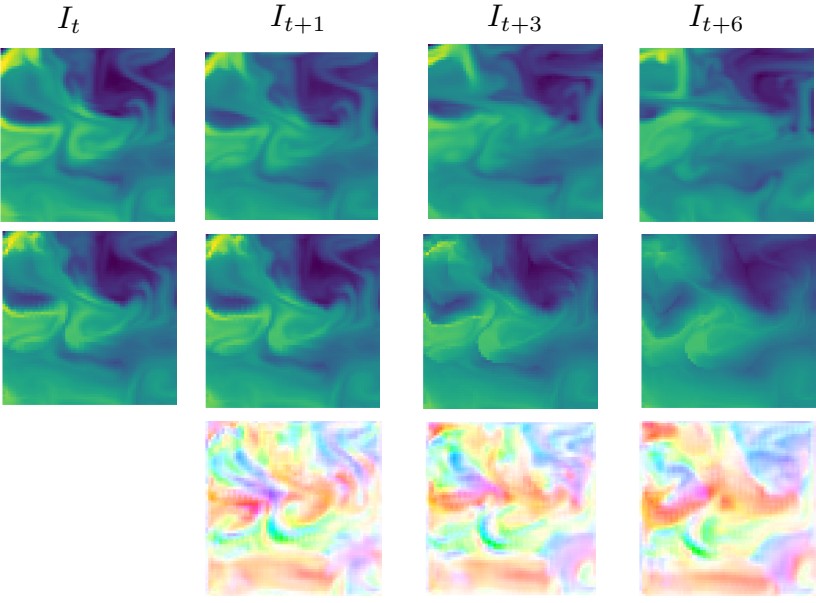

Figure 8: Output for the 6 of January to the 9 of January 2016. From top to bottom: target, our model prediction, our model flow

