# OpenReview forum: "Deep Learning for Physical Processes: Incorporating Prior Scientific Knowledge"
_ICLR.cc/2018/Conference — Accept (Poster)_

### Official Review · AnonReviewer1 · 2017-11-26
**The proposed method is interesting and offers nice perspectives for assimilation method. It opens interesting questions.**

**Rating:** 7
**Confidence:** 3

**Review:**

The paper ‘Deep learning for Physical Process: incorporating prior physical knowledge’ proposes
to question the use of data-intensive strategies such as deep learning in solving physical
inverse problems that are traditionally solved through assimilation strategies. They notably show
how physical priors on a given phenomenon can be incorporated in the learning process and propose
an application on the problem of estimating sea surface temperature directly from a given
collection of satellite images.

All in all the paper is very clear and interesting. The results obtained on the considered problem
are clearly of great interest, especially when compared to state-of-the-art assimilation strategies
such as the one of Béréziat. While the learning architecture is not original in itself, it is
shown that a proper physical regularization greatly improves the performance. For these reasons I
believe the paper has sufficient merits to be published at ICLR. That being said, I believe that
some discussions could strengthen the paper:
 - Most classical variational assimilation schemes are stochastic in nature, notably by incorporating
uncertainties in the observation or physical evolution models. It is still unclear how those uncertainties
can be integrated in the model;
 - Assimilation methods are usually independent of the type of data at hand. It is not clear how the model
learnt on one particular type of data transpose to other data sequences. Notably, the question of transfer
and generalization is of high relevance here. Does the learnt model performs well on other dataset (for instance,
acquired on a different region or at a distant time). I believe this type of issue has to be examinated
for this type of approach to be widely use in inverse physical problems.

---

> ### Author Response · Authors · 2017-12-28
> **Answers**
>
>
> Thanks a lot for your comments and suggestions.
>
> 1. Incorporating uncertainty in the model
> This is the next step of our work. We could introduce uncertainties in different forms. We started to work on a variant of this model using a scheme similar to conditional VAE (Variational Auto Encoder) with the idea that the model should be able to predict multiple potential vector flow candidates instead of a mean value. VAEs allow sampling from noise distributions and then generating diverse candidates. We have added a paragraph in the “Conclusion and future work” section where we discuss this point and provide some references indicating what type of approach could be used.
>
>
> 2. Generalization to other instances
>
>
> We agree that the potential of the model should also be evaluated for other conditions and at other places. This is however a whole study by itself involving dealing with different datasets, and performing many different types of tests. For now we do not have the availability of such datasets and this is left for further study.
> In order to provide some indications on the generalization performance, we however performed some additional tests on the available data. We evaluated the model on data distant in time and in space. For the former, we trained the model on the period 2011-2017 and tested on 2006-2010. The regions are the same as the one used in the main text (regions numbered 17 to 20 on figure 4). We have plotted the daily MSE on Figure 6 appendix B in the new version of the paper. The conclusion is that the range of error remains the same and there is a slight tendency for the error to increase when the time distance between test and train increases too.
>  For the latter (sequences from different regions), we have trained the model on 2 regions and tested on 2 other regions (and permuted the couples of regions). The two couples of regions have different dynamics. Results are provided on table 2, appendix B. The conclusion here is again that the range of error values remains the same. The error depends more on the region dynamics than on the train / test conditions. For regions with high dynamics, the loss is higher than for stable regions. Performance degrades more for the former regions than for the latter when the training set is sampled from a different region. Extensive additional tests should be performed in order to go beyond these partial conclusions.
>
> Note also that it is possible to fine-tune the model using available data of the specific zone in question. Other work, such as [Fischer and al], or [Ilg and al], suggest that deep models trained to predict a motion vector field can generalize from synthetic to real data, and when fine-tuned, there is an improvement in performance.
>
> [Fischer and al]: https://arxiv.org/abs/1504.06852
> [Ilg and al]: https://arxiv.org/abs/1612.01925

---

### Official Review · AnonReviewer2 · 2017-11-27
**interesting connection**

**Rating:** 7
**Confidence:** 3

**Review:**

In this paper, the authors show how a Deep Learning model for sea surface temperature prediction can be designed to incorporate the classical advection diffusion model. The architecture includes a differentiable warping scheme which allows back propagation of the error and is inspired by the fundamental solution of the PDE model. They evaluate the suggested model on synthetic data and outperform the current state of the art in terms of accuracy.

pros
- the paper is written in a clear and concise manner
- it suggests an interesting connection between a traditional model and Deep Learning techniques
- in the experiments they trained the network on 64 x 64 patches and achieved convincing results

cons
- please provide the value of the diffusion coefficient for the sake of reproducibility
- medium resolution of the resulting prediction


I enjoyed reading this paper and would like it to be accepted.

minor comments:
- on page five in the last paragraph there is a left parenthesis missing in the inline formula nabla dot w_t(x))^2.
- on page nine in the last paragraph there is the word 'flow' missing: '.. estimating the optical [!] between 2 [!] images.'
- in the introduction (page two) the authors refer to SST prediction as a 'relatively complex physical modeling problem', whereas in the conclusion (page ten) it is referred to as 'a problem of intermediate complexity'. This seems to be inconsistent.

---

> ### Author Response · Authors · 2017-12-28
> **Answers**
>
> Thanks a lot for the suggestions and comment, we corrected the mistakes in the updated version.
>
> The value of the diffusion coefficient in this case is 0.45, we have precised it in the updated version.
> We chose this value  of the image resolution in order to limit the complexity of the computations, but the model could be used as well with larger images.

---

### Official Review · AnonReviewer3 · 2017-12-04
**Learning the dynamics of an advection-diffusion equation from satellite images using CNNs, with a little help from then mechanistic model.**

**Rating:** 6
**Confidence:** 2

**Review:**

The authors use deep learning to learn a surrogate model for the motion vector in the advection-diffusion equation that they use to forecast sea surface temperature. In particular, they use a CNN encoder-decoder to learn a motion field, and a warping function from the last component to provide forecasting.

I like the idea of using deep learning for physical equations. I would like to see a description of the algorithm with the pseudo-code in order to understand the flow of the method. I got confused at several points because it was not clear what was exactly being estimated with the CNN. Having an algorithmic environment would make the description easier. I know that authors are going to publish the code, but this is not enough at this point of the revision.

Physical processes in Machine learning have been studied from the perspective of Gaussian processes. Just to mention a couple of references “Linear latent force models using Gaussian processes” and "Numerical Gaussian Processes for Time-dependent and Non-linear Partial Differential Equations"

In Theorem 2, do you need to care about boundary conditions for your equation? I didn’t see any mention to those in the definition for I(x,t). You only mention initial conditions. How do you estimate the diffusion parameter D? Are you assuming isotropic diffusion? Is that realistic? Can you provide more details about how you run the data assimilation model in the experiments? Did you use your own code?

---

> ### Author Response · Authors · 2017-12-28
> **Answers**
>
> Thanks a lot for your comments.
>
> 1. As you mentioned, the model is composed of two components: a CDNN which acts as a motion estimator, and a warping mechanism which predicts the future observation by moving the present observation along the motion field. The output of the CDNN is a vector field, i.e. a tensor of size WxHx2 where W and H are the width and height of the input images, and ‘2’ corresponds to the velocity.
>
> Inference and training work as follows:
>
> Inference - Input:  an image sequence (I_t-k+1,...,I_t) of k consecutive images representing temperature acquisitions. Output: an image sequence ( \hat{I}_t+1, \hat{I}_t+K) of K consecutive image predictions. Given the inputs (I_t-k+1,...,I_t) the CDNN will compute an estimated vector field \hat{w}_t. This vector field is then used to advect image I_t so as to compute an estimate of the future observation I_t+1. For multiple time step prediction (i.e. K > 1), the computed output \hat{I}_t+1 is fed back into the CDNN, leading to the following input sequence (I_t-k+2, …, I_t, \hat{I}_t+1) for estimating motion w_t+1. One can then estimate I_t+2 by advecting image \hat{I}_t+1 using \hat{w}_t+1, and so on.
>
> Training - The training set is a consecutive sequence of images.  An example is sampled from the training set and a loss value is computed between the model prediction and the target. Since the warping scheme (the solution to the advection-diffusion equation) is entirely differentiable, the gradient of the loss can be backpropagated through this component for modifying the parameters the CDNN module.
>
> Below is a pseudo-code for the training step
>
> Input: training set : sequence of SST images I_{1:T}
> Output: trained model parameters (CDNN parameters)
> Iterate until convergence
> --  Sample a sequence I_{t- k+1 : t+K} of images
>
> -- Forward pass of the model: using I_{t-k+1 : t }, we infer K future observations I_{t+1 : t+K} using the inference scheme proposed above.
>
> -- Compute the loss between the targets and model predictions.
>
> -- Backward pass of the model.  The gradient of the loss function with respect to the CDNN parameters is back propagated through the warping scheme in order to update the CDNN parameters via SGD (in the experiments we used Adam).
>
> 2. Thanks for the references on Gaussian processes. We did not go through the GP literature on the topic of physico-statistical modeling. Even if the methods and the application are different, the motivation and arguments are clearly similar.  We have added the references you suggested in the new version of the paper plus additional references in the related work section, under ‘ML for Physical modeling’.
>
> 3. In the theorem, a sufficient condition for the existence of this solution of the advection-diffusion equation, is that the image function I is square-integrable, (the square of I is Lebesgue integrable). A consequence is that I tends to zero as x approaches infinity. This will allow us to calculate the solution. In practice, we do consider that I has a compact support, i.e. I is zero outside its definition domain \Omega. This latter case is more specific than the square integrable hypothesis, and the theorem still applies. We have added a more detailed proof of the theorem in the appendix.
>
> 4. We did make the hypothesis that diffusion is isotropic and this is one of the simplification hypotheses adopted in the paper. Our intention was to give a proof of concept about the incorporation of prior knowledge, and to show that the proposed approach performed on par with more complex state of the art assimilation methods. If we focused on the application itself, improvements could probably be brought by including additional priors. In particular, attention mechanisms could be added to the warping mechanism for modeling anisotropy.
>
> 5. The diffusion parameter D is estimated on a validation set and its value is 0.45 - this is now indicated in the paper.
>
> 6. The data assimilation code is run  with a specific code provided by the authors of (Bereziat 2015). We had several interactions with the paper authors while performing the tests with their methods.

---

### Public Comment · (anonymous) · 2017-11-03
**technical questions**

Hi,

interesting paper. Could you please clarify the few details below?

1. In the proof of the theorem, how exactly did you obtain the fourier transform of the advection term of the PDE?
2. you mention that the input to the network is a "sequence of k consecutive SST images": what value of "k" did you use?
3. When applying the warping scheme, what is the policy for when the previous position "x - w" falls outside of the boundaries of the images?
4. Still in the warping scheme, how do you estimate the diffusion coefficient D?

Thank you and apologies if the answers are in the paper and I missed them.

---

> ### Author Response · Authors · 2017-11-07
> **Answer to technical questions**
>
> Thank you for the questions and remarks.
>
> 1. Since the integration is on R^2, we introduce the two corresponding 1-D variables, we then separate the Fourier transform of the advection term into two terms corresponding to each variable. Using an integration by parts for both terms and re-integrating with respect to the other variable, we obtain the solution. We will provide the details in a updated version.
> 2. The parameter k was chosen by testing on the validation set. The optimal tradeoff between complexity and accuracy appears to be k=4.
> 3. If the value of the pixel comes from outside the frame, we use the mean value of the pixels of the image. Since this is not informative, the model thus learns not to use values coming from outside the frame.
> 4. The diffusion coefficient is set as an hyperpameter.

---

### Public Comment · (anonymous) · 2017-11-17
**Interesting paper**

Nice work, really happy to see combining physics with DL does give such nice results.
I was wondering if you could answer a few questions:
1. In the regularization scheme - equation 6 - you are using the gradient and the divergence. With respect to what quantity (e.g. the input?) are you taking the gradient, and if it is the input, what is exactly is the divergence than when the input is not the cartesian coordinates but numerical values?

2. Could you elaborate in more detail of how "exactly" did you use NEMO in order to obtain the data used for training? My understanding is that you used the NOAA6 data and assimilate it into the NEMO engine. However, there are a few core engines under the NEMO project. I know this might be frustrating but for reproducibility, this would be very helpful as I don't think many people are familiar with the software. Or of how to actually to obtain the data from NOAA6.

---

> ### Author Response · Authors · 2017-11-19
> **Answer**
>
> 1. Gradient and divergence are computed wrt to the spatial variables x and y. To be more precise :
>
>  \left \| \nabla w \right \|^2 = (\frac{\partial u }{\partial x})^2 + (\frac{\partial u }{\partial y})^2 + (\frac{\partial v }{\partial x})^2 + (\frac{\partial v }{\partial y})^2
>
> and
>
> (\nabla . w)^2 = ( \frac{\partial u}{\partial x} + \frac{\partial v}{\partial y})^2
>
> 2.      We used directly the data generated by NEMO and available at
>
> http://marine.copernicus.eu/services-portfolio/access-to-products/?option=com_csw&view=details&product_id=GLOBAL_ANALYSIS_FORECAST_PHY_001_024

---

> > ### Public Comment · (anonymous) · 2017-11-22
> > **Thanks for the reply**
> >
> > Just for completeness, do you approximate the derivatives via numerical difference, given that you are predicting over a discrete set of coordinates? E.g.:
> > \frac{\partial u }{\partial x} \approx \sum_{i,j \in Dom} \frac{u(i,j) - u(i-1,j)}{\delta}

---

> > > ### Author Response · Authors · 2017-11-23
> > > **Reply**
> > >
> > > To approximate the values of the spatial derivatives, we use the forward Euler method, i.e. \frac{\partial u}{\partial x} \approx \frac{u_{n+1, k} - u_{n, k}}{\Delta x}, were x is the spatial discretisation value, and n and k are the spatial indices of the pixels.

---

### Author Response · Authors · 2018-01-04
**Summary of the corrections in the updated version of  the paper :**

- Minor corrections and typos, as suggested by the reviewers. This includes imprecision and missing references.

- A new paragraph has been included in the section "conclusion and future directions" for answering the comment of reviewer 2 regarding the modeling of uncertainty in the proposed model.

- A more detailed proof of our theorem 1 as been added in Appendix A.

- Some additional experiments were added in Appendix B, for answering (partially) the questions of reviewer 1 regarding the capacity of the model to generalize to new situations.

---

### Public Comment · (anonymous) · 2018-02-10
**Code**

Really nice work.
In the paper, you stated that the code would be made available upon publication.
Given that the final decisions are out and the paper is accepted could you open source your implementation?

---

### Author Response · Authors · 2018-04-30
**Code release**

Please checkout our implementation at https://github.com/emited/flow

---

### Decision · Program_Chairs · 2018-01-29
**ICLR 2018 Conference Acceptance Decision**

**Decision:**

Accept (Poster)

**Comment:**

This paper proposes to use data-driven deep convolutional architectures for modeling advection diffusion. It is well motivated and comes with convincing numerical experiments.
Reviewers agreed that this is a worthy contribution to ICLR with the potential to trigger further research in the interplay between deep learning and physics.